# Vaping Education: A Two-Year Study Examining Health Literacy and Behaviors in a Southeastern State

**DOI:** 10.3390/ijerph22071086

**Published:** 2025-07-08

**Authors:** Adrienne M. Duke-Marks, James Benjamin Hinnant, Jessica R. Norton, Linda M. Gibson-Young

**Affiliations:** 1Human Development and Family Sciences, College of Human Sciences, Auburn University, Auburn, AL 36849, USA; amd0046@auburn.edu (A.M.D.-M.); jbh0020@auburn.edu (J.B.H.); jrn0010@auburn.edu (J.R.N.); 2Nursing, College of Nursing, Auburn University, Auburn, AL 36849, USA

**Keywords:** vaping prevention, school-based program, adolescent health, electronic cigarettes, tobacco prevention

## Abstract

Electronic nicotine delivery systems (vapes) are the most used nicotine products among U.S. adolescents, with usage increasing significantly from 2017 to 2019. School-based prevention programs are a critical strategy for curbing youth vaping. This study utilized a retrospective pre/post survey to evaluate the effectiveness of a two-year school-based vaping prevention program utilizing a condensed version of the Stanford University Tobacco Prevention Toolkit. The program was implemented in-person and online across two years in a southeastern U.S. state. In year one, evalua-tion data were collected from 4252 youths from 20 rural counties who completed the in-person survey during the 2018–2019 program year. In year two, 1347 youths from 13 rural and urban counties completed the survey during the program year of 2019–2020. The key findings indicate significant increases in knowledge about vaping risks post-program. The findings from year one indicate that increases in knowledge about e-cigarettes were negatively related to the frequency of vaping, but this was not replicated in year two. Moreover, knowledge did not influence vaping frequency if youths had already started vaping, while pre-program knowledge did not predict the frequency of vaping in either year. These results suggest that vaping prevention education outcomes among youths are mixed.

## 1. Introduction

Electronic nicotine delivery devices (vapes) have become the predominant nicotine product among U.S. adolescents, with increasing prevalence in recent years. Despite the perception that vapes are less harmful than traditional cigarettes, research links vaping to adverse lung health, cognitive impacts, and an increased susceptibility to nicotine addiction. Given these risks, school-based prevention programs offer a strategic intervention to educate youths and mitigate vaping initiation. This manuscript will evaluate the effectiveness of a two-year school-based vaping prevention program utilizing a condensed version of the Stanford University Tobacco Prevention Toolkit. Implemented across 33 counties in a southeastern U.S. state, the program was delivered in-person and online.

Currently, electronic nicotine delivery devices, commonly known as vapes, are the most used nicotine product among U.S. adolescents [1]. National data show that from 2017 to 2019, past 30-day vape use increased from 11.7% to 27.5% among high school students and from 4.8% to 10.5% among middle school students [2]. Although youths commonly perceive vapes as being safer than traditional cigarettes, recent studies have shown that vape use compromises lung health [3,4], impacts the developing brain [5], and increases nicotine addiction and poly-substance use at an early age [6]. Vaping has also been linked to adverse mental health outcomes among adolescents, including depression and anxiety [7]. Educating youths on the risks of vaping may reduce the prevalence rates and risks associated with vaping.

### 1.1. Vaping Prevention Programs

Prevention programs can play an important role in curbing the growth of vaping by educating and informing youths of the risks and teaching refusal skills. Schools offer an important setting to do this work. School-based prevention programs have been cited as ideal contexts because they provide an opportunity to reach large numbers of adolescents in a focused learning environment [8].

Given the prevalence of vaping among adolescents, school-based prevention programs have been developed and evaluated. The findings from a recent review show a significant variation in the content, structure, and results of vaping prevention programs [9]. Programs typically focus on helping youths understand the health effects of vaping, the chemicals in flavors, and refusal skills [9]. There are a few school-based programs that have been evaluated with promising results, particularly the vaping modules from the Standford University Toolkit [10]. The Stanford University Tobacco Prevention Toolkit has lessons focused on the best practices for vape prevention, nicotine addiction, and health risks, as well as media literacy around marketing that targets youths [11]. Scholars have implemented vaping modules within the Prevention Toolkit in the form of multiple sessions, as well as a one-time, 30 min compressed version of the program [10,12]. Evaluation results are promising as there were increases in knowledge about e-cigarettes for both middle and high school youths, increases in youths’ perceptions of harm and addiction, and a lowered intent to start vaping [10]. Recent studies have also shown that utilizing theory is beneficial to understand the knowledge gains for youths and how it may translate into reducing vaping initiation [13].

The theory of planned behavior (TPB) can offer a theoretical framework for evaluating vaping prevention programs by addressing the cognitive and social determinants of adolescent behavior. The theory posits that behavior is primarily guided by intention, which is influenced by attitudes toward behavior, subjective norms, and perceived behavioral control [14]. In the context of vaping prevention, programs that effectively target adolescents’ negative attitudes toward vaping, correct misperceptions about peer approval, and build self-efficacy to resist peer pressure are more likely to reduce the intention to vape [15]. For example, programs incorporating TPB constructs have shown promise in modifying youth perceptions about the social acceptability and perceived risk of vaping, leading to significant reductions in vaping intentions [16].

### 1.2. The Current Study

We use a two-study design to examine two consecutive years of using a condensed version of the Stanford University Prevention Toolkit. We utilized three lessons and activities from the original version of the Toolkit. We used “Unit 3: What is so bad about E-cigarettes/Vape pens”, which explains nicotine addiction; “Unit 4: Why do e-cigarettes and vape pens matter to young people” to help students understand marketing strategies that seek to manipulate them into smoking; and “Unit 6: What are JUULs and other pod-based systems” to help youths understand the components and nicotine content of JUULs and pod-based systems, as well as the health risks associated with smoking. The learning objectives were to help youths understand vaping devices, their contents, and the associated health risks, as well as to help youths gain awareness of the strategies that manufacturers and marketing agencies use to increase vaping among youths.

Data from this study were collected across two years with in-person and online implementation and data collection options. We hypothesize that the program will still be effective in several ways. First, we hypothesize that in both years, health literacy related to vaping will change during the prevention program (post-test knowledge–pre-test knowledge). Second, we hypothesize that gains in health literacy pre- to post-program would be negatively related to vape use, regardless of race, sex, grade, and pre-program knowledge. Finally, we hypothesize that the relationship between change in health literacy and vape use will be moderated by pre-program knowledge such that the relationship between change in health literacy and vaping would be stronger for those with less pre-program knowledge.

## 2. Materials and Methods

### 2.1. Study Design and Participants

#### 2.1.1. Multi-Study Data Analysis Plan

The data collected in this study represent two distinct programming years. In year one, the program was implemented in person and utilized a paper evaluation survey. In year two, the program was implemented online and in-person and utilized an online survey. For each Study, univariate descriptive statistics and characteristics of the variables were evaluated, and then bivariate correlations were calculated as preliminary analyses (Table 1). To assess whether knowledge about vaping changed during the program, the mean differences in knowledge (post-test knowledge—pre-test knowledge) were evaluated with dependent samples *t*-tests for the survey items, as well as the average across items pre- and post-test. We used regression analysis to test hypothesis two and three, whether pre-post change in knowledge about vaping predicted vaping frequency in the past 30 days, controlling for race, sex, grade, and pre-test knowledge (addressing hypothesis two) and whether the relationship between change in knowledge and vaping frequency in the past 30 days depends upon pre-test knowledge (i.e., an interaction), controlling for race, sex, and grade (addressing hypothesis three).

Since vaping frequency in the past 30 days is an (approximate) count variable but indexed as an uncommon behavior (see Preliminary Analyses, below), zero-inflated Poisson (ZIP) regression was used to address the second and third hypothesis. The ZIP regression models two outcomes, the frequency data via Poisson regression and the latent zero-inflated membership process via logit regression and is thus a more appropriate and rigorous analytic choice relative to other regression choices in this case (e.g., simple linear regression) [17]. Maximum likelihood estimation with Monte Carlo integration was used to handle missing data, and robust standard errors were estimated for model parameters.

In all inferential statistical models, sex was dummy coded as 0 or 1 (female or male, respectively), and race was dummy coded with two variables (“Black or White” and “Black or Other”), with Black as the comparison group. To facilitate interpretation of the intercept estimates, all predictor variables were grand mean centered. In the case of a significant interaction, simple slopes for the interaction were calculated at conditional values (−1 *SD*, the mean, and +1 *SD*) of the moderator, pre-test knowledge, and plotted at these values. Results are presented separately.

#### 2.1.2. Study Sample

In year one, 4252 youths from 20 rural counties completed the in-person survey during the 2018–2019 program year. The sample was 50.9% male and 49.1% female. Racial categories included 54.4% European American/White, 29.7% African American/Black, 6.2% Hispanic Americans, 1.8% American Indian or Alaskan Native, 7.2% identified as being more than one race, and 0.7% of other races. Grades 4th through 12th were represented in the sample: less than 1% (*n* = 39) were in the 4th grade, 19.5% (*n* = 830) were in the 5–6th grades, 27.7% (*n* = 1177) were in the 7–8th grades, 38.4% (*n* = 1631) were in the 9–10th grades, and 13.5% (*n* = 575) were in 11–12th grades.

In year two, 1347 youths from 13 rural and urban counties completed an online retrospective pre-post survey in the program year of 2019–2020. Youth participants were 49.4% (*n* = 666) male and 46.7% (*n* = 615) female, with 3.8% (*n* = 51) who preferred not to respond. Youths described their race as White 49.6% (*n* = 668); African American/Black 33.5% (*n* = 451); More than one race 8.5% (*n* = 115); Latino/Hispanic 4.2% (*n* = 57); American Indian/Alaska Native 2.4% (*n* = 33); Asian 1.0% (*n* = 13); Native Hawaiian/Pacific Islander 0.3% (*n* = 4). Grade-levels spanned from 5th to 12th grade with 25.7% (*n* = 346) of youths in the 5–6th grade; 53.5% (*n* = 720) of youths were in the 7–8th grade; 14% (*n* = 188) of youths were in the 9–10th grade; and 6.9% (*n* = 93) of youths were in the 11–12th grade.

### 2.2. Program

The program used three lessons from the Stanford Tobacco Prevention Toolkit, covering nicotine addiction, industry marketing strategies, and health risks of vaping. The program was delivered in schools by trained Cooperative Extension educators. Schools were not selected based on specific criteria; rather, participation was contingent on their willingness to allow educators to implement the vaping prevention program with their students. The in-person program was delivered over three consecutive weeks, with one-hour sessions each week. Each session consisted of a PowerPoint presentation that incorporated interactive activities and allotted time for questions and discussion. The online version of the program was implemented synchronously via Zoom, during a single one-hour session. The same PowerPoint presentation was used, with integrated Kahoot! Quizzes to enhance engagement.

### 2.3. Measures

Demographic measures were used, and participants were asked their sex, race/ethnicity, and grade level.

#### 2.3.1. Health Literacy

Health literacy is a significant predictor of youth health behaviors. Specifically, youth health literacy measures the abilities, skills, commitments, and knowledge that contribute to their health-promoting decisions and actions [18,19,20]. In the context of health promotion programs, it encompasses various dimensions, including knowledge and self-efficacy [18]. Knowledge is considered an essential dimension of health literacy, encompassing facts, ideas, and situations related to health behaviors and practices. Self-efficacy is a foundational outcome in health literacy, as it measures the ability to engage in healthy behaviors [18,19].

There were seven items that assessed students’ knowledge of the ingredients of JUULpods and other vaping devices, as well as the effects of using these devices. Items include “Understanding how nicotine affects my brain”, “Knowledge that E-Cigarettes can contain nicotine”, “Knowledge that one JUULpod has as much nicotine as one pack of cigarettes”, “Knowledge of the long-term effects of vaping and JUULing”, “Knowledge of what’s in a JUULpod”, “How e-cig, vape, JUULpod manufactures target young people”, and “Confidence to avoid nicotine products (cigarette, smokeless tobacco, e-cigs, hookah, vapes, JUULpods)”. Many of the questions focused on JUUL pods, instead of general e-cigarettes because they were the most popular vaping devices for youths at that time [21]. These items were rated using a 4-point Likert scale ranging from 1 (*none*) to 4 (*A lot*).

#### 2.3.2. Electronic-Cigarette Use

Electronic cigarette use was measured by reporting the number of days they vaped in the last 30 days (*0 days*, *1–5 days*, *6–11 days*, *12–19 days*, and *20–30 days*).

### 2.4. Statistical Analysis

Dependent samples *t*-tests assessed pre- to post-program changes in vaping knowledge. Zero-inflated Poisson (ZIP) regression modeled vaping behavior, controlling for demographics and pre-program knowledge levels.

#### Study Procedures

Surveys were completed at the end of the program. In year one, a paper survey was completed. In year two, a survey link was placed in the Zoom chat and a QR code was placed at the end of the PowerPoint. Regardless of whether the test was conducted in-person or online, youths completed a retrospective pre-post survey; they responded twice to each question, first noting their knowledge level before the program, and then identifying their current knowledge after participating in the program. We used retrospective pre-post surveys instead of pre/posttest to reduce financial and time allocations, while allowing us to still measure knowledge gains [22,23]. The retrospective scores are found to be comparable to true pretest and posttest data in determining the extent of a program’s impact [24]. No identifying information was collected.

## 3. Results

### 3.1. Changes in Knowledge

Significant increases in vaping-related knowledge were observed in both study years (*p* < 0.001). Year 1 participants demonstrated larger knowledge gains compared to Year 2 participants.

### 3.2. Predictors of Vaping Behavior

Regression analyses revealed that knowledge gains were significantly associated with lower likelihood of vaping (*p* < 0.01). However, the relationship between knowledge gains and vaping frequency was inconsistent across the two study years.

#### 3.2.1. Change in Knowledge About Vaping

In year 1, dependent samples *t*-tests indicated significant mean differences in youth-reported knowledge about vaping pre- and post-program on all eight items, as well as the average pre- and post-knowledge scores (Table 2). Mean differences ranged from 0.46 (*SD* = 0.94; *t* (4294) = 31.71, *p* < 0.001) to 1.35 (*SD* = 1.27; *t* (4274) = 69.82, *p* < 0.001) and the average pre-post difference was 0.99 (*SD* = 0.80; *t* (3820) = 76.58, *p* < 0.001), indicating a significant increase in knowledge pre- to post-program.

In year 2, dependent samples *t*-tests showed significant mean differences in pre-post program knowledge for all eight items and the average pre-post difference (Table 2). In this sample, mean differences were smaller than those seen in Study 1 and ranged from 0.24 (*SD* = 0.86; *t* (1346) = 10.26, *p* < 0.001) to 0.85 (*SD* = 1.10; *t* (1346) = 28.30, *p* < 0.001) and the average pre-post difference was 0.54 (*SD* = 0.67; *t* (1346) = 29.26, *p* < 0.001), indicating a significant increase in knowledge pre- to post-program

#### 3.2.2. Prediction of Vaping Behavior in the Past 30 Days

In relation to predicting count data under the Poisson distribution, White and Other youths (compared to Black youth) exhibited higher levels of vaping behavior. Grade was positively related to vaping behavior with older youths in higher grades exhibiting more vaping behaviors. Pre-program knowledge was not related to vaping behavior, but pre-post change in knowledge was negatively related to vaping behavior (i.e., youths who had greater increases in knowledge about vaping had lower levels of vaping behavior; *B* = −0.38, *p* < 0.01). In predicting zero-inflation membership, grade was negatively related to zero-inflation (i.e., youths in earlier grades were less likely to have vaped). Pre-program knowledge was positively related to zero-inflation membership, indicating that youths with a greater pre-program knowledge about vaping were less likely to have vaped (*B* = 0.40, odds = 1.49, *p* < 0.01). Similarly, changes in knowledge pre- to post-program were positively related to the zero-inflation of vaping membership (i.e., youths who had greater increases in knowledge about vaping were less likely to have vaped; *B* = 0.34, odds = 1.40, *p* < 0.01). The interaction between pre-program knowledge and change in knowledge did not predict zero-inflated membership.

In year 2, none of the variables predicted count data of vaping behaviors. Grade was negatively related to zero-inflation of vaping membership (i.e., youths in earlier grades were less likely to have vaped). Additionally, pre-program knowledge and pre-post change in knowledge were positively related to zero-inflation of vaping membership (i.e., youths who had higher pre-program knowledge about vaping or who had greater increases in knowledge about vaping were less likely to have vaped; *B* = 0.88, odds = 2.41, *p* < 0.01 and *B* = 1.11, odds = 3.03, *p* < 0.01, respectively). The interaction between pre-program knowledge and change in knowledge did not predict zero-inflated membership. The model results for both years are presented in Table 3.

## 4. Discussion

The findings support the effectiveness of school-based vaping prevention programs in increasing health literacy. The results underscore the importance of early prevention efforts of targeting middle school students before vaping behaviors become established. While health literacy was associated with reduced initiation, intervention programs are needed to impact vaping frequency.

The evaluation examined pre- to post-program changes in health literacy and found consistent evidence of a program effect, thus supporting hypothesis 1. We also investigated associations between pre-program health literacy and changes in knowledge regarding the contents and health risks of vaping with 30-day e-cigarette use. Across the two years, findings indicated that higher baseline knowledge and those with greater increases in knowledge pre- to post-program were associated with a lower likelihood of e-cigarette use. These results suggest that increased knowledge may play a meaningful role in shaping adolescents’ attitudes toward vaping, a key construct of the theory of planned behavior. According to TPB, behavioral intention is influenced by attitudes, perceived social norms, and perceived behavioral control. By enhancing knowledge of the risk and chemical contents of e-cigarettes, the program may have contributed to more negative attitudes toward vaping and possible greater perceived behavioral control to resist. Notably, despite the variations in delivery format, there were similar results.

Evidence for associations between base-line health literacy or change in health literacy and the frequency of vaping was less compelling. Findings from year one indicates that increases in health literacy concerning vaping were negatively related to the frequency of vaping; however, this was not replicated in year two. Pre-program health literacy did not predict the frequency of vaping in either study. Thus, hypothesis 2 received consistent support for categorical vaping, but less compelling evidence in regard to the frequency of vaping. Hypothesis 3, which was a conditional protection-enhancing effect for youths with less pre-intervention knowledge in the change in health literacy to e-cigarette usage association, was not supported in either study. Unfortunately, we do not have data to capture actual initiation timing in relation to the program to contextualize our findings more clearly.

Although e-cigarette use was low for most youth, usage did appear to increase with grade level. This finding is like other studies examining vaping and age or grade [25,26]. Pre-adolescent youths have very low vaping rates; therefore, a focus on grades 4th through 6th may be best for prevention programs. The purpose of universal prevention programs is to educate youths about the risk of electronic cigarettes before the initiation of vaping behaviors. If youths can obtain research-based information before being influenced by other sources of information that may be incorrect, prevention efforts may be more effective.

When exploring other studies, our findings are consistent with other prevention work using the e-cigarette modules from the Stanford University Tobacco Toolkit [10,27] prevention program. Educating youths before they start vaping is important to decreasing vaping behaviors. Youths who are knowledgeable about the risks associated with vaping may be more likely to make informed decisions in the future [28]. Knowledge may prevent youths from initiating the behavior, but as the results show, knowledge does not influence vaping frequency once youths start vaping. Therefore, knowledge of vaping risk before initiation is important. Although knowledge gains do not always translate into behavioral modification, knowledge of risk is an important first step toward preventing e-cigarette use. More research on e-cigarette prevention programs is critical to understanding ways to reduce the number of youths who start using these devices.

There are several limitations in this study. The online data collected was from a single-session program, and neither year had a control group, resulting in a lack of causal inference and limited opportunities to report change. Furthermore, self-reported, retrospective pre/post data collection can impact the reliability of the data collected. In particular, participants may reinterpret their previous knowledge or attitudes based on newly acquired perspectives, impacting the accuracy of pre-program self-assessment. Similarly, retrospective designs rely on participants’ ability to recall past perceptions and knowledge, which can be limited or distorted. Lastly, social desirability bias is always a potential limitation with self-reported data, particularly with youths. However, in an online environment, there may be enough perceived anonymity and reduced peer influence to reduce this type of bias.

## 5. Conclusions

This study adds to the growing evidence supporting school-based vaping prevention programs. Our results indicate that more targeted efforts are still needed before disinformation by the nicotine industry can be thwarted. Future research should explore long-term behavioral effects and optimal program delivery methods to maximize impact.

## Figures and Tables

**Table 1 ijerph-22-01086-t001:** Descriptive statistics and correlations in studies 1 and 2.

**Study 1**							
	**1**	**2**	**3**	**4**	**5**	**6**	**7**
1. Days Vaped	-						
2. Pre-Knowledge	0.10 **	-					
3. Change in Knowledge	−0.13 **	−0.83 **	-				
4. Race: White vs. Black	0.17 **	0.13 **	−0.11 **	-			
5. Race: Other vs. Black	−0.05 **	−0.03	0.04 *	−0.47 **	-		
6. Gender	0.03	0.04 *	−0.08 **	−0.02	−0.00	-	
7. Grade	0.21 **	0.28 **	−0.25 **	0.15 **	−0.11 **	−0.01	-
** *N* **	4312	3875	3663	4263	4263	4254	4240
** *M* **	0.37	2.74	0.99	0.54	0.16	0.50	2.43
** *SD* **	0.96	0.79	0.80	0.50	0.37	0.50	0.98
**Study 2**							
	**1**	**2**	**3**	**4**	**5**	**6**	**7**
1. Days Vaped	-						
2. Pre-Knowledge	−0.01	-					
3. Change in Knowledge	−0.15 **	−0.68 **	-				
4. Race: White vs. Black	−0.08 **	0.00	0.07 *	-			
5. Race: Other vs. Black	0.08 **	0.05 *	−0.10 **	−0.44 **	-		
6. Gender	0.02	0.05	−0.09 **	−0.04	0.09 **	-	
7. Grade	0.09 **	0.00	−0.01	0.01	−0.06 *	−0.05	-
** *N* **	1347	1347	1347	1341	1341	1281	1347
** *M* **	0.21	3.09	0.54	0.50	0.17	0.52	2.02
** *SD* **	0.74	0.66	0.68	0.50	0.37	0.50	0.82

*Note:* * *p* < 0.05. ** *p* < 0.01.

**Table 2 ijerph-22-01086-t002:** Dependent samples *t*-tests of average pre- to post-knowledge change in studies 1 and 2.

	*M* _Pre_	*SD* _Pre_	*M* _Post_	*SD* _Post_	Δ Knowledge	*T* Value (*df*)	*p*
Study 1	2.74	0.78	3.73	0.46	0.99	76.58 (3820)	<0.001
Study 2	3.09	0.66	3.63	0.54	0.54	29.26 (1346)	<0.001

**Table 3 ijerph-22-01086-t003:** Zero-inflated Poisson (ZIP) regression model estimates for study 1 and study 2 in relation to vaping use.

	Study 1				Study 2			
Poisson regression	*B*	95% *CI*	*β*	95% *CI*	*B*	95% *CI*	*β*	95% *CI*
Intercept	0.30	[0.16, 0.41]	-	-	0.46	[0.26, 0.66]	-	-
Race (B, W)	0.31 *	[0.05, 0.57]	0.42	[0.07, 0.77]	−0.32	[−0.71, 0.08]	−0.70	[−1.26, −0.14]
Race (B, O)	0.43 **	[0.15, 0.72]	0.43	[0.16, 0.69]	−0.01	[−0.47, 0.45]	−0.02	[−0.77, 0.74]
Sex (F, M)	0.04	[−0.10, 0.18]	0.06	[−0.13, 0.24]	0.17	[−0.14, 0.49]	0.38	[−0.31, 1.07]
School Grade	0.19 **	[0.07, 31]	0.51	[0.24, 0.77]	0.12	[−0.07, 0.31]	0.43	[−0.18, 1.03]
Pre-Knowledge	−0.11	[−0.32, 0.09]	−0.24	[−0.69, 0.22]	0.06	[−0.18, 0.31]	0.18	[−0.50, 0.87]
Δ Knowledge	−0.38 ***	[−0.57, −0.20]	−0.83	[−1.25, −0.40]	−0.08	[−0.33, 0.17]	−0.24	[−1.03, 0.55]
Pre × Δ Knowledge	−0.05	[−0.21, 0.11]	−0.10	[−0.41, 0.22]	−0.08	[−0.28, 0.13]	−0.20	[−0.77, 0.36]
Zero-inflated membership	*B*	95% *CI*	*β*	95% *CI*	*B*	95% *CI*	*β*	95% *CI*
Intercept	1.45	[1.30, 1.58]	-	-	2.14	[1.88, 2.40]	-	-
Race (B, W)	−0.96 ***	[−1.26, −0.67]	−0.24	[−0.32, −0.17]	0.03	[−0.48, 0.54]	0.01	[−0.13, 0.14]
Race (B, O)	−0.14	[−0.51, 0.24]	−0.03	[−0.10, 0.05]	−0.30	[−0.87, 0.27]	−0.06	[−0.17, 0.05]
Sex (F, M)	−0.15	[−0.35, 0.04]	−0.04	[−0.09, 0.01]	0.27	[−0.17, 0.71]	0.07	[−0.04, 0.18]
School Grade	−0.55 ***	[−0.68, −0.42]	−0.28	[−0.34, −0.22]	−0.33 **	[−0.55, −0.12]	−0.14	[−0.23, −0.05]
Pre-Knowledge	0.40 **	[0.15, 0.65]	0.16	[0.06, 0.26]	0.88 ***	[0.46, 1.30]	0.30	[0.16, 0.44]
Δ Knowledge	0.34 **	[0.11, 0.57]	0.14	[0.05, 0.23]	1.11 ***	[0.59, 1.63]	0.39	[0.22, 0.56]
Pre × Δ Knowledge	0.07	[−0.11, 0.25]	0.03	[−0.04, 0.09]	0.08	[−0.37, 0.53]	0.02	[−0.12, 0.17]
AIC	54,668.49				15,825.96			
BIC	54,995.59				16,091.45			

*Note:* * *p* < 0.05. ** *p* < 0.01. *** *p* < 0.001.

## Data Availability

The datasets presented in this article are not readily available because of privacy protections.

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
