# Peer review of "Vaping Education: A Two-Year Study Examining Health Literacy and Behaviors in a Southeastern State"

_ijerph, 2025, doi:10.3390/ijerph22071086_

Round 1
Reviewer 1 Report
Comments and Suggestions for Authors
This retrospective, cross-sectional study evaluates answers on knowledge (health risks of ecigs) and on use (frequency in the past 30 days) given by 4,252 students from 20 rural counties in Alabama during an intervention program 2018-2019 ('study 1') and by 1,347 students from 13 urban and rural counties in a pre-post survey in the program year of 2019-2020 ('study 2'). As expected, increases in knowledge about vaping risks post-intervention were significant. Adolecents who had greater knowledge before intervention or larger increases in knowledge about vaping from pre- to post-Intervention were less likely to have used ecigs. However, a protection-enhancing effect for youth with less pre-intervention knowledge in the change in knowledge to e-cigarette usage association, was not supported, indicating that more targeted efforts are still needed before disinformation by the nicotine industry can trigger initiation. This could be added to conclusions, together with the limitations of possible selection bias from participation. The important result that knowledge did not influence vaping frequency once youth started vaping, should definitely be added to the abstract.
Author Response
Please see the attached response to reviewers. We appreciate you allowing our team to review according to reviewer comments.
Thank you,
Linda Gibson-Young

Reviewer 2 Report
Comments and Suggestions for Authors
This manuscript evaluates a two-year school-based vaping prevention program using a condensed version of the Stanford University Tobacco Prevention Toolkit across 33 counties in a southeastern U.S. state. The study demonstrates significant increases in vaping-related knowledge following the intervention and shows associations between knowledge gains and reduced likelihood of vaping initiation.
I have some comments:
Abstract
- The abstract should mention the use of retrospective pre-post surveys.
- Add sample sizes for both studies directly into the abstract.
Introduction
- Consider incorporating a behavioral change theory (e.g., Health Belief Model, Theory of Planned Behavior) to better contextualize the knowledge-behavior relationship
- Sections 1.2 ("The Current Studies") and 1.3 ("Multi-Study Data Analysis Plan") contain methodological content that would be more appropriately placed in the Methods section rather than the Introduction. I recommend relocating these sections to the Methods section to improve the manuscript's structure and alignment with standard scientific reporting guidelines.
Methods
- (optional) The two-study design is a bit confusing as presented. I think it would be better to restructure as a single study with two implementation years rather than "Study 1" and "Study 2," which implies separate investigations.
- While the authors cite literature supporting this approach, they should more thoroughly discuss its limitations, particularly:
- Response shift bias
- Social desirability effects
- Memory recall issues
- How these might differentially affect the two study years
- The knowledge scale includes confidence to avoid nicotine products. Discuss whether this is conceptually a knowledge or self-efficacy item.
Results
- Clarify the magnitude of knowledge change (effect sizes) to give a sense of practical significance.
Discussion
- Connect findings to behavioral change theory and explain the knowledge-behavior relationship mechanisms
- Expand Limitations Section to include:
- Retrospective pre-post design limitations
- Lack of control group
- Self-report bias and potential for unmeasured confounding.
- The relatively short-term assessment.
- The inability to capture actual initiation timing relative to the intervention
Author Response

(The authors gave the same response as above.)

Round 2
Reviewer 2 Report
Comments and Suggestions for Authors
Authors addressed previous comments.
No further comments.